# Recursive Objective Space Exploration (ROSE): A computationally efficient deterministic approach for bi-objective optimization

**Ihab Hashem, Viviane De Buck, Seppe Seghers, Jan Van Impe** [ID]*

Chemical Engineering Department, KU Leuven, BioTeC & OPTEC, Ghent, Belgium

* jan.vanimpe@kuleuven.be

## Abstract

Bi-objective optimization problems arise when a process needs to be optimized with respect to two conflicting objectives. Solving such problems produces a set of points called the Pareto front, where no objective can be improved without worsening at least one other objective. Existing deterministic methods for solving Multi-Objective Optimization Problems (MOOPs) include scalarization techniques, which transform the problem into a set of Single-Objective Optimization Problems (SOOPs) where each of them is to be solved independently to obtain a point on the Pareto front. In this paper, we propose an alternative strategy that tackles bi-objective optimization problems by exploring the objective space recursively at a reduced computational cost. Our approach is inspired by how plants efficiently explore physical space in search of light energy, balancing exploration and exploitation while minimizing biomass cost. The algorithm navigates the objective space without revisiting previously explored areas by solving intermediate SOOPs visualized as branches within this space. A trade-offs based stopping criterion enables the algorithm to focus on steep, information-rich segments of the Pareto front, creating denser branches to provide more detailed representation of the Pareto front. We benchmark the algorithm's performance against a standard scalarization-based solution strategy from the literature, employing five case studies. Our strategy demonstrates that a holistic approach that structures the solution process within the objective space provides significant advantages. These include a more computationally efficient method for solving bi-objective optimization problems, an adaptive representation of the Pareto front based on trade-offs, and intuitive, straightforward parameterization guided by a user-oriented, trade-offs based stopping criterion.

## 1 Introduction

Solving a MOOP involves identifying points within the objective space where the value of any objective cannot be improved without worsening at least one other objective. This set of

**Data availability statement:** The underlying code and case studies are publicly available through the KU Leuven Research Data Repository at https://doi.org/10.48804/NGXLMW.

**Funding:** This research was supported by the Fund for Scientific Research-Flanders (FWO) under project G.0B41.21N. The funder's website is https://www.fwo.be. IH, JVI received support from this funding. The funders had no role in the study design, data collection and analysis, decision to publish, or preparation of the manuscript.

**Competing interests:** The authors have declared that no competing interests exist.

points is known as the Pareto front. In practice, solution algorithms aim to find an approximation of the Pareto front, by producing a finite discrete set of points. This set is then provided to a Decision Maker (DM), who can be either human or automated, for consideration. The DM then selects the solution that best suits their needs [31,41]. Multi-objective optimization continues to play a key role in real-world applications ranging from healthcare operations and supply chain logistics to IoT-enabled environmental monitoring [1,2,24].

Generally, two classes of algorithms are used to solve MOOPs: vectorization and scalarization-based methods [28]. Vectorization algorithms work stochastically and directly tackle the MOOP by generating a set of solutions in the objective space that gradually converge towards the Pareto front [5,12]. On the other hand, scalarization methods transform the MOOP into a set of Single-Objective Optimization Problems (SOOPs) using a parameterization scheme. Afterwards, each SOOP is independently solved to yield a point on the Pareto front [30,32].

Recent progress in the field of multi-objective optimization has seen significant developments across various approaches. In the area of vectorization algorithms, [45] enhanced the Multi-Objective Evolutionary Algorithm based on Decomposition (MOEA/D) with a new method for adjusting weight vectors via linear interpolation, addressing discontinuous Pareto fronts in bi-objective optimization. [43] combined the Hypervolume Newton Method (HVN) with a multi-objective evolutionary algorithm to tackle constrained MOPs. Additionally, [4] applied a Multi-Objective Imperialist Competitive Algorithm (MOICA) to solve a hub location-routing bi-objective optimization problem. Surrogate-assisted evolutionary algorithms have also been developed to reduce the cost of expensive multi-objective optimization [23,26,38]. With respect to scalarization methods, [19] introduced a novel scalarization technique, the unified direction method, extending the Pascoletti–Serafini approach. [3] proposed an exact scalarization method for bi-objective integer linear optimization, utilizing diverse reference points instead of traditional weighting factors. [27] applied the weighted Tchebycheff optimization method within an uncertainty-based framework for multi-objective optimization. In the domain of multi-objective decision-making, strides have been made in aiding decision-makers in selecting solutions from the Pareto set. [42] introduced the Deep Ranking Analysis by Power Eigenvectors (DRAPE) to tailor solution rankings according to the decision-maker's preferences by assigning weights to different criteria. [40] proposed the hierarchical version of the Best-Worst Method (HBWM) for decision-making processes involving multiple criteria and sub-criteria.

Both vectorization and scalarization-based algorithms face two major challenges. The first hurdle is that not all regions of the Pareto front are equally significant to the DM [29]. An efficient algorithm should prioritize providing a denser representation of the Pareto front's steeper segments where solutions have a high level of trade-offs between them. Meanwhile, low trade-offs segments or flat regions of the Pareto front can have a sparser representation [29]. Secondly, long computational times can be an issue when solving MOOPs [25]. While scalarization algorithms are generally less time-consuming than vectorization algorithms, converting a MOOP into a series of SOOPs incurs computational time that scales linearly with the number of SOOPs to be solved. This can be a significant bottleneck when solving large-scale problems [25].

Despite these advances, there remains a pressing need for algorithms that can efficiently allocate computational effort where it matters most, to provide detailed representations of the Pareto front in the regions that are relevant for decision making, while avoiding redundant calculations in uninformative areas. Furthermore, most existing methods lack an intuitive, problem-oriented stopping criterion, often forcing users to rely on arbitrary choices for solution density and termination. To address these challenges, we propose a new strategy and

apply it for solving bi-objective optimization problems. Our approach is based on the principle that the exploration of the objective space to find the Pareto front must be efficient: no part of the objective space should be explored more than once. To achieve this, we have implemented a branching strategy inspired by phototropism - the process by which the branches of a tree systematically explore physical space when growing towards a light source.

Phototropism is the process by which plants respond to sunlight. A signaling pathway is initiated upon sensing light, causing cells to grow towards its source. As the plant grows towards the light, it produces finer branches to fill any gaps. By having denser branching at light-rich regions and sparser branching at light-poor regions, the plant can capture as much energy as possible while reducing biomass production.

Similarly, our algorithm systematically explores the objective space by creating intermediate optimization problems. The solving process of these problems corresponds to branching within the objective space. This ensures an efficient exploration process of the objective space as no part of the objective space is explored more than once. Additionally, the algorithm structures the solution process such that later optimization problems, visualized as branches in the objective space, start from initial points that are progressively closer to the Pareto front. As a result, the Pareto points found later in the solution process are obtained at fewer iterations than the initial points.

The algorithm also incorporates a built-in criterion to terminate exploration of flat regions of the Pareto front. This leads to denser branching only at the steep, information-rich areas of the Pareto front. By doing so, our algorithm captures as much information as possible about the shape of the Pareto front while reducing computational costs. This results in a more informative Pareto set in fewer iterations than traditional scalarization methods.

Accordingly, our algorithm offers three key advantages: (i) a more efficient exploration of the objective space, resulting in a computationally efficient solution process with fewer iterations; (ii) an adaptive representation of the Pareto front based on trade-offs, featuring a higher density of points in regions with steep slopes that are of greater interest to decision-makers; and (iii) intuitive and straightforward parameterization, requiring only a user-oriented, trade-offs based stopping criterion and a single initial guess for the decision variables.

The paper is structured as follows: Sect 2 presents the mathematical formulation of MOOPs, as well as an overview of popular scalarization methods, namely Weighted Sum (WS) and Normal Boundary Intersection (NBI) methods, along with the posteriori analysis of a Pareto front. Sect 3 outlines the concept of the new algorithm for solving bi-objective optimization problems. Sect 4 provides an overview of the case studies used in this paper. The performance of the algorithm is benchmarked and illustrated via the case studies in Sect 5. Finally, Sect 6 summarizes the paper's conclusions.

## 2 Mathematical formulation

The formulation of a MOOP as a minimization problem is provided following [25]:

$$\min_{x}\{J_1(x), J_2(x), \dots, J_m(x)\} \tag{1}$$

subject to:

$$h(x) = 0 \tag{2}$$
$$g(x) \leq 0 \tag{3}$$
$$a \leq x \leq b \tag{4}$$

with $x$ the decision vector to be optimized. $J_i(x)$ represents the $i$th objective function and $m$ represents the number of objectives in the problem. The equality and inequality constraints are represented by $h(x)$ and $g(x)$ respectively. The lower and upper boundaries of $x$ are represented by $a$ and $b$ respectively.

The feasible solution set $S$ is defined as all vectors $x$ that satisfy the imposed constraints. A solution $x^*$ is deemed a Pareto optimal solution for the MOOP if there exists no other solution $x$ for which $J_i(x) < J_i(x^*)$ holds true for all objectives simultaneously, while $x^*$ has at least one objective $J_k$ that satisfies $J_k(x^*) < J_k(x)$.

Algorithms for solving MOOPs can be broadly classified into two classes: vectorization algorithms and scalarization algorithms [28]. Vectorization methods [12,35,39] tackle the problem by directly generating a set of solutions using a stochastic algorithm. This set of solutions gradually evolves in the objective space towards the Pareto front via consecutive rounds of mutation, crossover, and selection operations. Scalarization methods, on the other hand, are deterministic, gradient-based algorithms that work by discretizing the original MOOP into a set of SOOPs using a weight vector [25]. By varying the weight vector parameters, solving each sub-problem renders a corresponding point on the Pareto front. Scalarization methods are known for their ability, relative to vectorization algorithms, to handle large-scale problems with complex constraints, their higher computational efficiency, and their ability to work efficiently in high-dimensional space. However, they are more prone to converge to local optima compared to stochastic methods [25]. The trade-offs between the two classes of algorithms make each of them more suitable for tackling different types of problems [28]. In this paper, we focus on scalarization methods, with two classical algorithms, WS and NBI, being highlighted in this section.

## 2.1 Weighted Sum (WS)

The WS method is a popular technique for solving MOOPs, primarily due to its simplicity. It involves constructing a combined objective by assigning a weight to each objective. Solving the problem repeatedly while varying these weights leads to obtaining a distribution of points on the Pareto front. This method can be formulated as follows [25]:

$$\min_x \sum_{i=1}^{m} w_i J_i(x) = w^\top J(x) \tag{5}$$

The weight vector $w$ is composed of the set of weights $w_1, w_2, \dots, w_m$, such that $w_i \geq 0$ and $\sum_{i=1}^{m} w_i = 1$. Despite the simplicity of the WS method, it has two major drawbacks [9]. First, a uniform distribution of weights does not correspond to a uniform distribution of points on the Pareto set, meaning that the resulting Pareto set is not evenly distributed on the Pareto front. Second, it is not possible to obtain solutions in the convex segments of the Pareto front using this method. These drawbacks have led to the development of techniques that employ alternative parameterization schemes.

## 2.2 Normal Boundary Intersection (NBI)

The NBI method is a robust algorithm for solving MOOPs [10]. It is based on two concepts: the Utopia Point (UP) and the Convex Hull Of Individual Minima (CHIM). The UP is defined as the vector composed of all the individual minima of the different objectives. The method first finds the individual minima of the different objectives, i.e., the anchor points. The objectives of the problem are then re-scaled to shift the UP to the origin. The CHIM is a convex

                                                    

hull formed by connecting the anchor points; it is a hyperplane that is constructed in the objective space by connecting all the individual minima of the problem.

Afterwards, a set of parameterized SOOPs is formulated such that their objective functions are to maximize the lengths of a corresponding set of vectors that are quasi-normal to the CHIM and pointing towards the UP. Solving each of these SOOPs yields a point on the Pareto front. The normalization of the objectives, along with the usage of the CHIM, allows for obtaining a uniform distribution of points on the Pareto front, including a representation of the convex segments. This mitigates some of the issues encountered by the WS method [10].

However, several drawbacks remain associated with using NBI. For instance, there is no criterion for selecting the number of solutions a priori to obtain a distribution with a targeted density/resolution to satisfy the DM's needs [29]. Several modifications have been proposed to allow the NBI to provide a denser representation at the regions of interest to the DM. This can be achieved either by the interactive generation of solutions, leaving to the DM the decision of further exploring the interesting segments of the Pareto front [6,41], or by employing a recursive technique for generating the NBI points, wherein the solution process halts early at the flatter segments of the Pareto front [20]. In both scenarios, the objective space is suboptimally explored due to the discretization into SOOPs and the independent solution of each, leading to increased and unnecessary computational costs.

## 2.3 Posteriori analysis

After obtaining a set of Pareto points, the next step is typically to carry out a posteriori analysis of the Pareto set [15,22,34]. This involves filtering and ranking the solutions according to their perceived potential importance to the DM. An intuitive strategy is to only keep the solutions that have significant trade-offs among them. This can be done using the Smart Filter (SF) algorithm [29]. The algorithm works by making pairwise comparisons between the points in the Pareto set. The main parameter of the algorithm is $\Delta t$, which is defined as the trade-off level that is significant to the DM. If a Pareto point does not outperform existing points in the set for any objective by a margin greater than $\Delta t$, it is considered unnecessary for the DM and is excluded from the filtered set. The remaining points in the filtered Pareto set, known as the Smart Pareto set, would exhibit significant trade-offs among them, and hence they are relevant to a potential DM. The major drawback of this method is that it requires obtaining a dense Pareto set initially, which results in wasted computational effort to obtain solutions that will subsequently be filtered [20,29].

## 3 Algorithm description

### 3.1 Concept

Branching is a remarkable example of optimization in nature [21]. Trees have evolved over millions of years to capture more sunlight with less biomass, by growing branches to explore their environment efficiently; avoiding growing more branches where they are not needed. To achieve this, trees must find an optimal branching structure by balancing exploration and exploitation: first they explore space via sparser branching to capture more light and then exploit areas with high energy availability by producing denser branching there. The spatial distribution of the branches has to exhibit an optimal 'resolution' for efficient energy capture; a sparser branching structure than necessary captures insufficient amount of energy, while a denser structure translates into a higher cost of biomass that is not offset by the marginal excess in captured energy [21,33]. This phenomenon has similarities with solving MOOPs, where the goal is to obtain a set of solutions that provides sufficient information about the

Pareto front. Here also, the set has to have an optimal resolution: a too sparse representation fails to represent it correctly, while a too dense representation leads to extra computational effort for low gain in information. Hence, inspired by the branching phenomenon, we propose a novel Recursive Objective Space Exploration (ROSE) algorithm that efficiently explores the objective space by using a branching scheme.

The algorithm begins at an initial point in the objective space, referred to as the seed ($S$). $S$ corresponds to the initial guess point in the decision space, denoted as $x_0$ and provided by the user. The first step is similar to NBI; in a bi-objective 2D space, the algorithm solves two single objective optimization problems: $\min J_1$ and $\min J_2$ to obtain anchor points $A_2$ and $A_1$ respectively. The anchor points values are used to normalize the objective functions and shift the UP to the origin.

The three points: $S$, $A_1$, and $A_2$, shown in Fig 1, together define the growth region. The growth region represents the area of the objective space to be explored by the algorithm through a structured solution process, visualized by branches.

The algorithm constructs a branch in the objective space by solving a SOOP that is formulated as follows:

$$\max_x \left| \vec{B}_{A_1,A_2} \right| \tag{6}$$

with the vector $\vec{B}_{A_1,A_2}$ defined as:

$$\vec{B}_{A_1,A_2} = \left| \vec{SA_2} \right| \vec{SA_1} + \left| \vec{SA_1} \right| \vec{SA_2} \tag{7}$$

such that $\vec{B}_{A_1,A_2}$ originates from the seed and bisects the two vectors $\vec{SA_1}$ and $\vec{SA_2}$. Solving this optimization problem yields a point $P$ on the Pareto front (see Fig 1).

The algorithm then continues recursively by defining a new node, which serves as the starting point in the objective space for the new branches. The point in the objective space

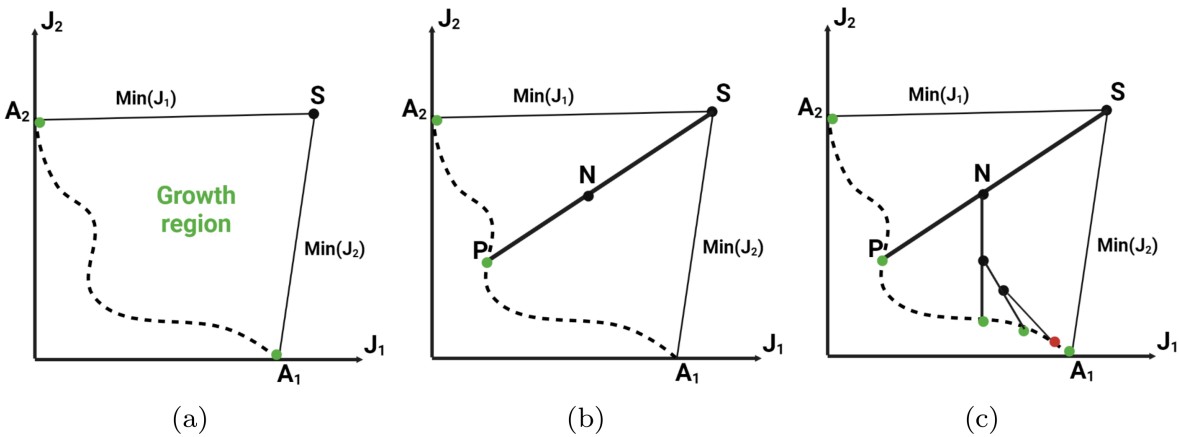

(a)  (b)  (c)

**Fig 1. Illustration of the recursive objective space exploration process implemented by the ROSE algorithm.** (a) The process starts from an initial guess (seed) $S$ and identifies the boundaries of the region to be explored by independently minimizing each objective to obtain the anchor points $A_1$ and $A_2$. (b) The algorithm then constructs an intermediate optimization problem, visualized as a branching operation: it seeks a new solution $P$ on the Pareto front by maximizing the length of a vector that bisects the directions from $S$ to $A_1$ and $S$ to $A_2$. This branching step is repeated recursively: (c) each new branch divides the region further, with new optimization problems constructed between existing solutions. Branch growth automatically stops in regions where further improvement offers only insignificant trade-offs (as controlled by $\Delta t$), leading to a denser set of solutions in steep, information-rich regions of the Pareto front, and a sparser set elsewhere. This strategy ensures efficient, adaptive exploration of the objective space without redundant computations.

closest to the center of the previous branch is selected to be the next node. Two additional SOOPs are constructed on each side of $\vec{B}_{A_1,A_2}$ to maximize the lengths of two new vectors bisecting $\{S\vec{A}_1, \vec{B}_{A_1,A_2}\}$, and $\{\vec{B}_{A_1,A_2}, S\vec{A}_2\}$. The algorithm continues recursively, generating more points on the Pareto front. When two Pareto points are found within the trade-off level $\Delta t$, defined a priori by the user, their corresponding branches inhibit the growth of each other, and no new branches are created between them. This results in a Pareto front with a predefined resolution $\Delta t$. The stopping criterion enables denser branching in regions with high trade-offs, high exploitation, while maintaining less dense branching in the low trade-offs regions of the Pareto front. An overview of the algorithm is presented in Algorithm 1.

**Algorithm 1 Recursive objective space exploration algorithm.**

```
Input: Initial guess x_o, and significant trade-off level Δt
```

```
Output: Pareto set with adaptive resolution S.
```

*Step 1:* Initialize the solution set $S = \{\}$.

*Step 2:* Generate the seed point based on $x_o$: $(J_1(x_o), J_2(x_o))$

*Step 3:* Solve $\min_x\{J_1\}$ and $\min_x\{J_2\}$ to get anchor points $A_2$ and $A_1$.

*Step 4:* Start the recursive process with $S$ as a branching node and $\{A_1, A_2\}$ as Pareto points.

 Construct vector $\vec{B}_{A_1,A_2}$ based on Equation 7.

 Solve sub-problem $\min_x -|\vec{B}_{A_1,A_2}|$ to find Pareto point $P_i$.

 Find the node $N_i$ closest to the branch's center.

 **if** $P_i$ and $A_1$ satisfy the *significance condition* **then**

 Call the recursive function with node $N_i$ and Pareto points $\{P_i, A_1\}$ as input.

 Set Flag $F_1$ to *True*.

 **if** $P_i$ and $A_2$ satisfy the *significance condition* **then**

 Call the recursive function with node $N_i$ and Pareto points $\{P_i, A_2\}$ as input.

 Set Flag $F_2$ to *True*.

 **if** $F_1$ and $F_2$ are *True* **then**

 Add $P_i$ to $S$

*Step 5:* When all recursive calls are exited, produce the solution set $S$.

The proposed approach saves computational cost in two ways. First, it systematically explores the objective space without revisiting any previously explored regions. Second, by coupling the algorithm with the stopping criterion using $\Delta t$, it ensures that branching occurs only in the relevant regions of the Pareto front, thereby saving computational effort. Furthermore, the approach provides an intuitive problem-based stopping criterion for solving a bi-objective optimization problem, unlike traditional methods that set an arbitrary number of solutions beforehand and need filtering afterwards.

Compared to generating the Pareto front using the NBI method followed by Smart Filter (NBI/SF), ROSE utilizes an adaptive strategy. Both approaches have computational costs that scale as $\mathcal{O}(N)$, where $N$ is the number of Pareto points at the chosen resolution. In NBI/SF, a fixed number $k$ of scalarized SOOPs are solved uniformly, resulting in a total cost of $\mathcal{O}(k)$, even though many solutions may be redundant and later filtered out. In contrast, ROSE generates new SOOPs only in regions where the local trade-off exceeds a threshold $\Delta t$, yielding $N_{\text{ROSE}} = \mathcal{O}(M)$, where $M \ll k$ in typical cases. In the worst case ($M = k$), both methods solve the same number of SOOPs. Additionally, ROSE reduces the average solver iterations per SOOP by warm-starting each subproblem from a nearby Pareto point. Let $I_{\text{NBI}}$ and $I_{\text{ROSE}}$

denote the average solver iterations per subproblem for NBI/SF and ROSE, respectively. Since $I_{ROSE} < I_{NBI}$ in practice, total costs become $T_{NBI} = \mathcal{O}(kI_{NBI})$ versus $T_{ROSE} = \mathcal{O}(MI_{ROSE})$, further compounding ROSE's efficiency advantage.

### 3.2 Software

MATLAB R2021a was used to implement the algorithm and solve the case studies tackled in this paper. The default interior-point algorithm in MATLAB's optimization toolbox, *fmincon*, was utilized to solve the SOOPs.

## 4 Case studies

The algorithm was benchmarked against a standard scalarization-based strategy, the NBI method [10], as referenced in literature [8,16,18,37]. The benchmarking involved three numerical bi-objective problems, an industrial case study on the bi-objective optimization of an I-Beam, and a numerical case study with three objectives to demonstrate the algorithm's applicability to higher-dimensional problems. The selection of these case studies aims to test the ROSE algorithm's capability in exploring diverse geometries of objective spaces and Pareto front shapes.

### 4.1 A numerical bi-objective problem

The first benchmark problem that is used in this paper is the numerical bi-objective problem (BIOBJ), formulated by [29] to illustrate the performance of the SF. The problem is defined as follows [29]:

$$\min_{x}\{x_1, x_2\} \tag{8}$$

subject to:

$$\left(\frac{x_1 - 10}{10}\right)^8 + \left(\frac{x_2 - 5}{5}\right)^8 - 1 \leq 0 \tag{9}$$

$$-10 \leq x_1 \leq 10 \tag{10}$$

$$-10 \leq x_2 \leq 10 \tag{11}$$

The Pareto front of this problem is characterized by a sharp knee and long plateaus. The desired performance of a MOO algorithm is hence to obtain a denser representation at the knee region that is characterized by a high level of trade-offs.

### 4.2 CONSTR problem

The CONSTR problem is formulated by [13] as follows:

$$\min_{x}\{J_1, J_2\} \tag{12}$$

subject to:

$$J_1 = x_1 \tag{13}$$

$$J_2 = (1 + x_2)/x_1 \tag{14}$$

$$g_1(x) = (9x_1 + x_2) \geq 6 \tag{15}$$

$$g_2(x) = (9x_1 - x_2) \geq 1 \tag{16}$$

$$x_1 \in [0.1, 1.0] \tag{17}$$

$$x_2 \in [0, 5.0] \tag{18}$$

The Pareto front of the CONSTR-problem is characterized by a sharp division between two segments: a steep section that with high level of a trade-offs between solutions and a flat section. A MOO algorithm must provide a denser representation of the steep segment.

### 4.3 DO2DK problem

The third benchmark problem aims to evaluate an algorithm's ability to produce a representation of a complex Pareto front with multiple knees. The problem is formulated by [7], building on earlier work by [11] and [14], as follows:

$$\min_x \{J_1, J_2\} \tag{19}$$

with the objectives defined as follows:

$$J_1(x) = g(x)r(x_1)\left(\sin(\pi x_1/2^{s+1} + (1 + \frac{2^s - 1}{2^{s+2}}\pi) + 1\right) \tag{20}$$

$$J_2(x) = g(x)r(x_1)\left(\cos(\pi x_1/2 + \pi) + 1\right) \tag{21}$$

subject to:

$$g(x) = 1 + \frac{9}{n-1}\sum_{i=2}^{n} x_i \tag{22}$$

$$r(x_1) = 5 + 10(x_1 - 0.5)^2 + \frac{1}{k}\cos(2k\pi x_1)2^{s/2} \tag{23}$$

$$0 \leq x_i \leq 1, \quad i = 1, 2, ..., n \tag{24}$$

This formulation enables the manipulation of the Pareto front's number of knees and skewness through the parameters $k$ and $s$, respectively, where $n$ represents the number of decision variables in the problem.

### 4.4 I-Beam optimization

The fourth problem is the optimization of an I-Beam, proposed by [44], where the two objectives $J_1$ and $J_2$ are the minimization of the weight of the beam and maximizing its displacement respectively. The problem is formulated as follows [17]:

$$\min_x \{J_1, J_2\} \tag{25}$$

with the objectives defined as follows:

$$J_1(x) = 2x_1x_2 + x_3(x_1 - 2x_4) \tag{26}$$

$$J_2(x) = \frac{PL^3}{48EI(x)} \tag{27}$$

subject to:

$$g(x) = \frac{M_y}{W_y(x)} + \frac{M_z}{W_z(x)} - K_g \tag{28}$$

$$x_1 \in [10, 80] \tag{29}$$

$$x_2 \in [10, 50] \tag{30}$$

$$x_3 \in [0.9, 5.0] \tag{31}$$

$$x_4 \in [0.9, 5.0] \tag{32}$$

with:

$$Wy(x) = \frac{2x_2 x_4 \left(4x_4^2 + 3x_1 (x_1 - 2x_4)\right) + x_3 (x_1 - x_4)^3}{6x_1} \tag{33}$$

$$Wz(x) = \frac{2x_4 x_2^3 + x_3^3 (x_1 - x_4)}{6x_2} \tag{34}$$

$$I(x) = \frac{2x_2 x_4 \left(4x_4^2 + 3x_1 (x_1 - 2x_4)\right) + x_3 (x_1 - 2x_4)^3}{12} \tag{35}$$

with $P = 600kN$ and $Q = 50kN$ representing orthogonal and cross-sectional forces respectively. $L = 200cm$ denotes the length of the beam. $K_g = 16kN/cm^2$ and $E = 2 \times 10^4 kN/cm^2$ correspond to the stress yield and Young's modulus respectively. $M_y = 30000kNcm$ and $M_z = 2500kNcm$ represent the maximum bending moments around the x and y axes respectively.

In addition, the formulation of a problem with three objectives is given below to explore the algorithm's capacity for extension to higher dimensions.

## 4.5 A numerical 3 objectives problem

This problem is adopted from [36] and formulated as follows:

$$\min_x \{J_1, J_2, J_3\} \tag{36}$$

subject to:

$$J_i = x_i, \quad i = 1, 2, 3 \tag{37}$$

$$0.2 \le x_i \le 10, \quad i = 1, 2, 3 \tag{38}$$

$$x_1 \ge x_2^{-1} + x_3^{-1} \tag{39}$$

$$x_2 \ge x_1^{-1} + x_3^{-1} \tag{40}$$

$$x_3 \ge x_1^{-1} + x_2^{-1} \tag{41}$$

## 5 Results and discussion

In this section, we evaluate the performance of the ROSE algorithm using the case studies introduced earlier and compare it to a standard solution strategy from literature. First, we measure the computational speed of the algorithm, expressed by the average number of iterations required to obtain a solution on the Pareto front, and compare it with the NBI method for obtaining a uniform representation of the Pareto set consisting of the same number of solutions for both methods, using the BIOBJ problem as a case study. Second, we assess the

ability of the algorithm to provide a representation of a Pareto front with adaptive resolution, along with its speed, and compare it with the standard strategy of posteriori filtering of a Pareto front produced by the NBI method using the SF algorithm for all three case studies.

## 5.1 BiObjective problem

First, the ROSE algorithm's ability to produce a uniform representation of the Pareto front, with no consideration to the trade-offs levels discrepancies between different segments, is tested and benchmarked against the NBI method. The main goal of this test is to see if the improved exploration strategy of the branching algorithm will lead to more efficient exploration of the objective space that would manifest itself in lower number of iterations required by the solver per Pareto point produced. Both algorithms are tested on the BIOBJ problem using the same initial decision vector $x_0$. Fig 2 shows the Pareto sets obtained by both the ROSE algorithm and NBI method, as well as a visualization of the solution process of the ROSE algorithm in the objective space. In this figure, the size of the Pareto sets that have been obtained by both methods is equal to 65 solutions. With respect to the ROSE algorithm, this has been obtained by utilizing the number of recursion levels as a stopping criterion instead of $\Delta t$, which has been set to $\Delta t = 0$. For this run, the maximum number of recursion levels for the ROSE algorithm has been set to 6, corresponding to 65 points of the Pareto front. It is seen in this figure that the branching solution process of the ROSE algorithm, in absence of a trade-offs-based stopping criterion, has led to obtaining a uniform representation of the Pareto front, similar to the one obtained via NBI.

Nonetheless, the true advantage of the ROSE strategy, which can be observed by examining the evolution of the solution process in the objective space, is its systematic exploration of the objective space without revisiting any previously explored regions. It is important to note that each branch in this context represents the solution process of an intermediate SOOP, with the branch node being the point in the objective space that corresponds to the initial decision vector fed to the solver. As the solution process progresses, branches get initiated from nodes that lie closer to the Pareto front, leading to a lower number of iterations required to obtain a solution. In contrast, NBI, and most traditional scalarization methods, begin by

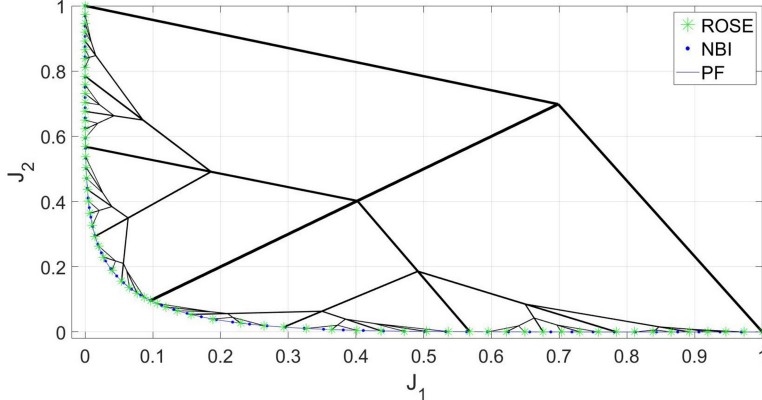

**Fig 2. Pareto front representations obtained by solving the BIOBJ problem using the ROSE algorithm and the NBI algorithm, as well as a visualization of the solution process of the ROSE algorithm in the objective space.** Both representations feature an equal number of points, 65 points, corresponding to 6 recursive levels of the ROSE algorithm.

scalarizing the MOOP into a series of SOOPs that are solved independently, potentially leading to repeated exploration of the objective space. In this example, the more efficient exploration strategy implemented in the ROSE algorithm has resulted in a 44% reduction in the average number of iterations required by the solver for each Pareto point found on the Pareto front. A comparison between the ROSE algorithm and NBI, regarding the average number of solver iterations per Pareto point across different numbers of Pareto points/ROSE recursion levels, can be seen in Fig 3. Notably, the ROSE algorithm needs fewer solver iterations per Pareto point as the number of Pareto points increases, unlike the NBI. This can be attributed to the branching scheme, where the branches leading to later points originate from nodes that are increasingly located closer to the Pareto front.

The other key advantage of the branching scheme is its compatibility with a trade-offs-based termination criterion. By setting a predefined significant trade-offs level value, denoted as $\Delta t$, a branch can be "inhibited" if it produces a Pareto point that falls within the vicinity of another branch. Inhibition, in this context, means that no new solution processes will originate from the inhibited branch, resulting in no further growth in the area between the two branches.

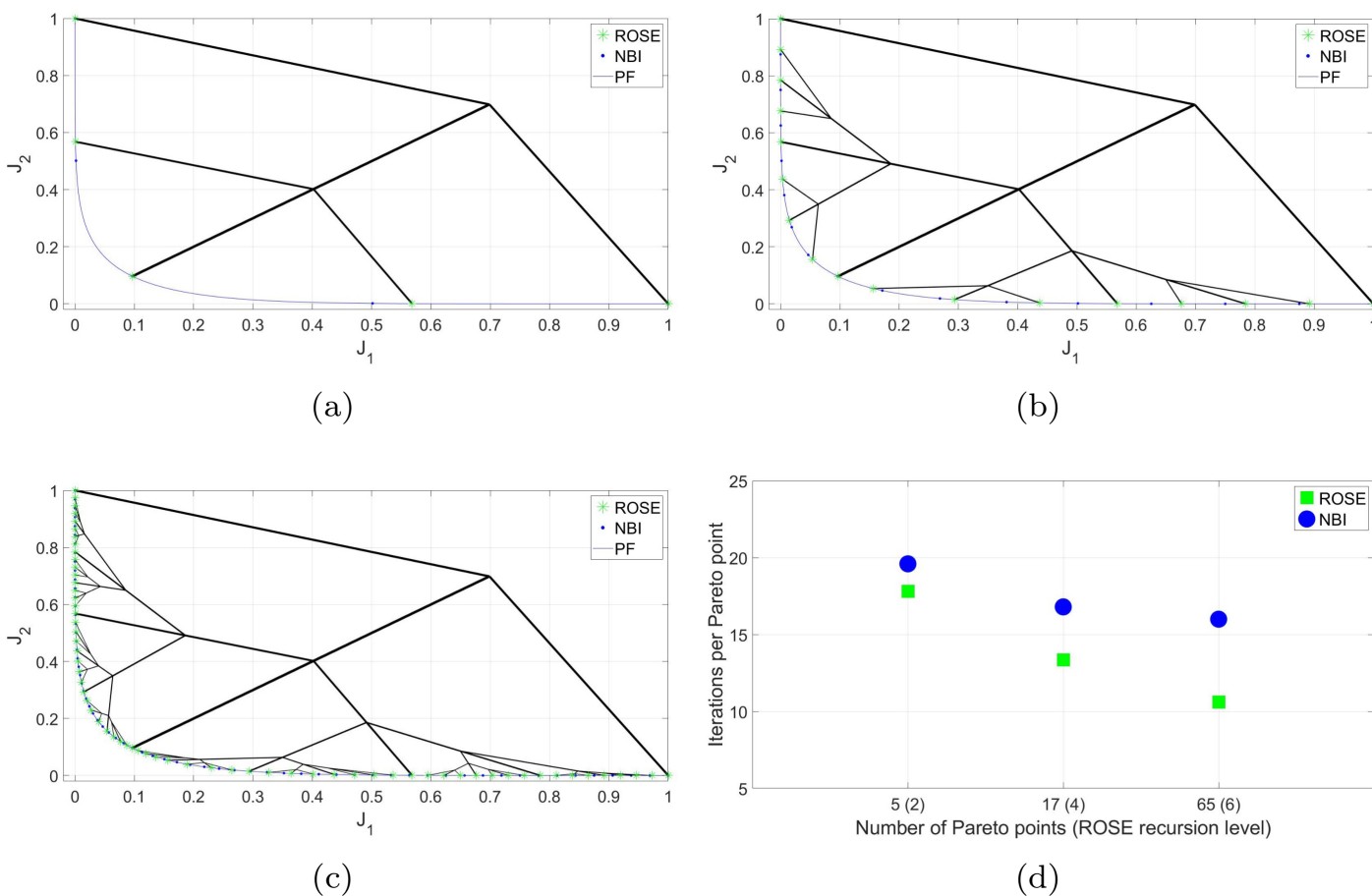

**Fig 3. Comparing ROSE and the NBI algorithms with varying sizes of the Pareto set/levels of recursion of the ROSE algorithm when applied to the BIOBJ problem: (a) 5 points/two recursion levels (b) 17 points/4 recursion levels (c) 65 points/6 recursion levels (d) Comparing mean number of iterations per Pareto point for both algorithms across different sizes of the Pareto set/recursion levels.**

As illustrated in Fig 4, when utilizing the ROSE algorithm with a trade-offs termination criterion of $\Delta t = 0.01$, in absence of any upper limit on the number of recursion levels, the algorithm's growth pattern varies depending on the explored region of the Pareto front. Sparse branching is observed around plateau regions, while denser branching and a higher level of exploitation occur in the information-rich knee regions of the Pareto front. This approach offers two benefits. Firstly, it provides a simpler and more intuitive stopping criterion compared to the combination of NBI with SF. In the latter approach, depicted in the same figure, an arbitrarily dense representation of the Pareto front must be obtained first using the NBI method, with no predefined method for specifying the desired number of points to be obtained by the algorithm. Subsequently, SF must be applied a posteriori with the level of significant trade-offs ($\Delta t$) specified as an input parameter. Secondly, the early inhibition of solution processes at the plateau regions would avoid the high computational overhead and excessive computation associated with the NBI/SF approach.

Therefore, although the Pareto set generated by the ROSE algorithm is similar to the one obtained using the NBI/SF approach in terms of size and spatial distribution, a significant difference in the number of iterations required to obtain this representation is observed. Fig 5 demonstrates the efficiency of the ROSE algorithm by comparing the average number of iterations per Pareto point produced against the NBI/SF approach. It is also noticed that the speed improvement increases more noticeably as $\Delta t$ decreases. This can be attributed to two factors. First, for the NBI/SF approach, a lower $\Delta t$ necessitates the generation of a denser Pareto front, resulting in more points being filtered and a higher number of iterations per Pareto point produced. Secondly, an ROSE algorithm operating with a low $\Delta t$ produces a finer representation of the Pareto set through branches originating in the objective space at locations steadily closer to the Pareto front. As a result, the number of iterations per Pareto point decreases as $\Delta t$ decreases and the number of generated points increases.

It is crucial to mention that the NBI here has been calibrated to yield the minimum number of points necessary for the SF to operate effectively. For every significant point in the filtered Pareto set, at least one insignificant point from the original set is eliminated. However, it is not feasible to guarantee this condition in advance. In real-world scenarios, more points might be generated, resulting in an even higher computational cost.

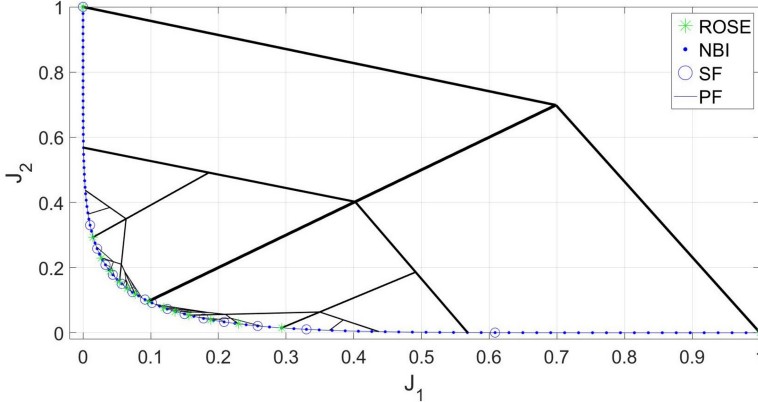

**Fig 4. Pareto front representations with adaptive density obtained by solving the BIOBJ problem using the ROSE algorithm and the NBI algorithm coupled with posteriori filtering via a SF are depicted, along with a visualization of the solution process of the ROSE algorithm in the objective space.** The stopping criterion of the ROSE algorithm, which aligns with the SF specification, was set to $\Delta t = 0.1$.

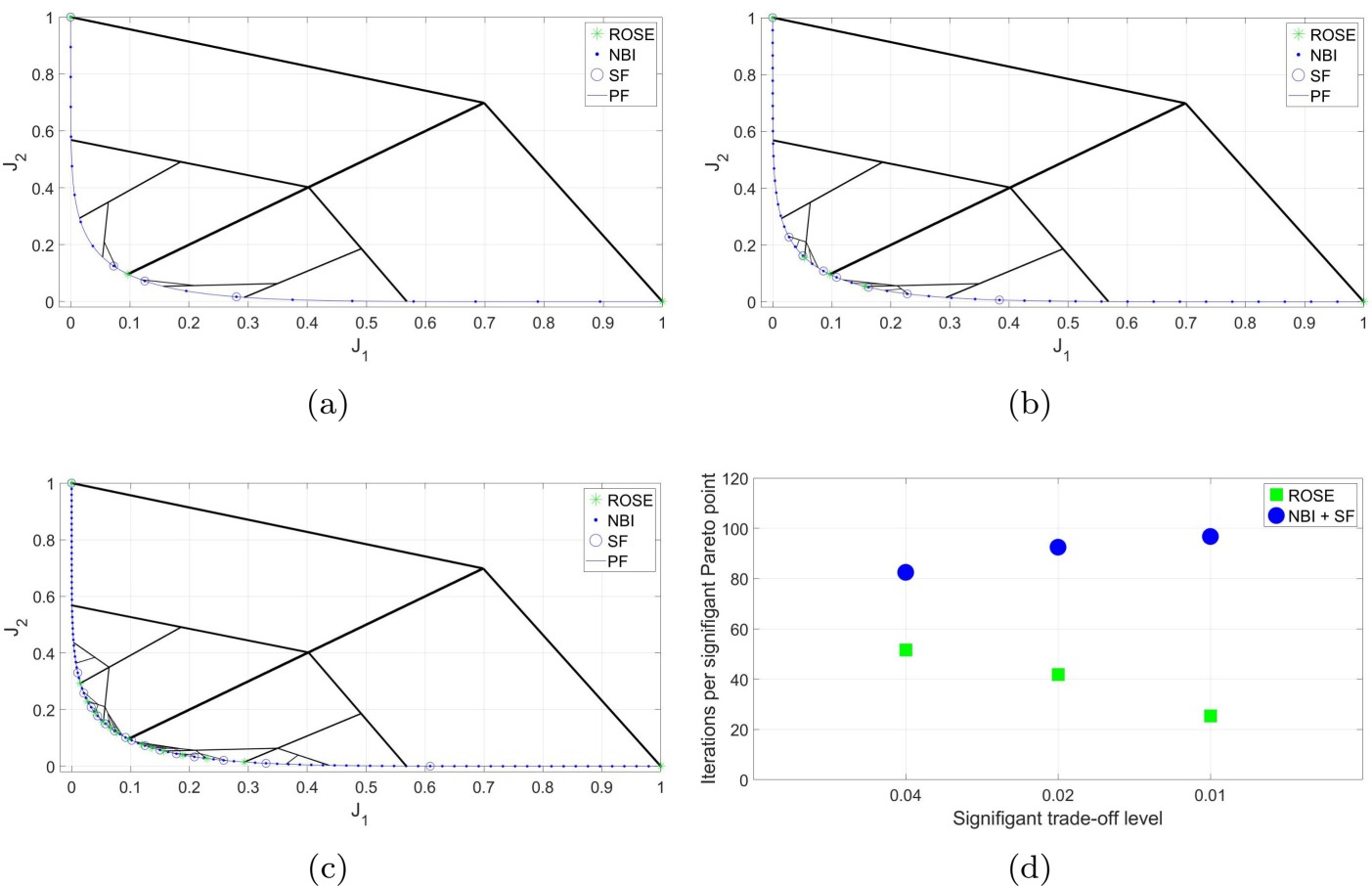

**Fig 5. Comparing ROSE and the NBI algorithm combined with a SF with varying resolutions of the ROSE stopping criterion and the SF specification when applied to the BIOBJ problem: (a)** $\Delta t = 0.04$ **(b)** $\Delta t = 0.02$ **(c)** $\Delta t = 0.01$ **(d) Comparing the mean number of iterations per Pareto point for both algorithms across different ROSE stopping criterion/SF specifications.**

## 5.2 CONSTR- problem

In Fig 6, the ROSE algorithm is utilized to achieve an adaptive resolution of the Pareto front for the CONSTR-problem. It is compared to the NBI/SF approach using a significant trade-off level stopping criterion of $\Delta t = 0.01$. The figure illustrates the solution process of the ROSE algorithm. Notably, the starting point in the objective space, the seed, is positioned non-centrally, in contrast to the previous example. However, the algorithm successfully obtains a comprehensive representation of the Pareto front with adaptive resolution; the density of the branches, and consequently the obtained Pareto points, is higher in the steep region of the Pareto front compared to the flat segment. The ability of the ROSE algorithm to quickly differentiate between these two types of segments cannot be replicated using the NBI/SF approach. In the NBI/SF approach, a dense representation is required for both the steep and the flat segments before the SF can be applied effectively.

The consequence of that is a discrepancy in the average number of iterations per Pareto point between the two approaches, see Fig 7. And similar to the previous case study, the trend of the higher gain in performance of ROSE compared to the NBI/SF approach as $\Delta t$ decreases holds on here as well.

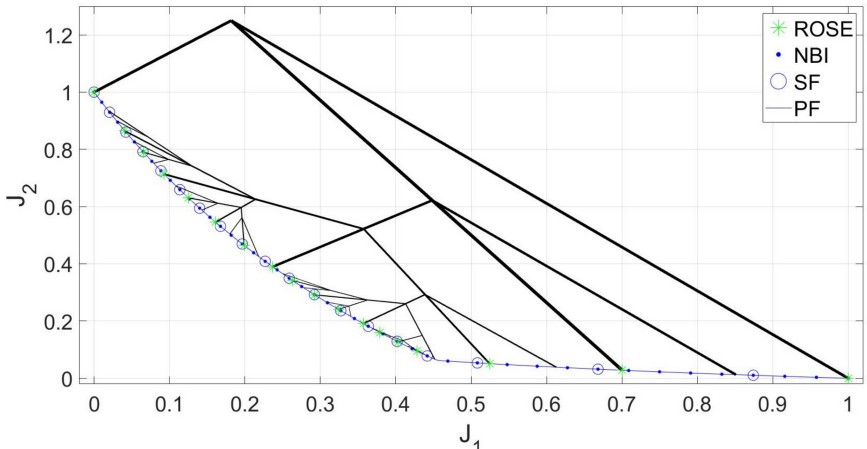

**Fig 6. Pareto front representations for with adaptive density obtained by solving the CONSTR- problem using the ROSE algorithm and the NBI algorithm coupled with posteriori filtering via a SF are depicted, along with a visualization of the solution process of the ROSE algorithm in the objective space.** The stopping criterion of the ROSE algorithm, which aligns with the SF specification, was set to $\Delta t = 0.01$.

## 5.3 DO2DK problem

The third case study is the DOD2K problem that is characterized by having a Pareto front that is challenging to represent, owing to its complex geometry and the existence of multiple knees. The problem's parameters were chosen to be $S = 2$ and $k = 3$. When applying the ROSE algorithm with $\Delta t = 0.003$, it was successfully able to locate all the knees of the Pareto front, and to represent them according to their trade-off level, as seen in Fig 8.

The comparison of the mean number of iterations per Pareto point between the ROSE algorithm and the NBI/SF approach is presented in Fig 9.

It is noted here that, as observed in Fig 8, the ROSE algorithm shares a pitfall of the NBI algorithm, namely the generation of non-Pareto points in the non-convex regions of the Pareto front. This is a problem that is typically solved by subsequently filtering the resulting set using a global Pareto filter [25].

**5.3.1 I-Beam optimization.** Both the ROSE algorithm and NBI/SF are applied to the optimization of the I-Beam, with the same desired resolution of $\Delta t = 0.005$. The problem was tackled multiple times using varying values of $x_o$. Modifying the initial guess of the solver corresponds to starting from different points in the objective space. As observed in Fig 10, both methodologies successfully represented the Pareto front with adaptive resolution with a higher density of points in the knee region of the Pareto front.

In line with previous results, the ROSE algorithm outperformed the NBI/SF method across all initial points. When analyzing the average number of iterations per significant Pareto point, as shown in Fig 10, the performance of the ROSE algorithm was less sensitive to the starting point within the objective space. In contrast, the NBI/SF method exhibited varying efficiencies depending on the value of the initial guess. This variability can be attributed to the fact that a disadvantageous guess might correspond to initiating from an unfavorable position within the solution space, resulting in slower convergence. Since all SOOPs in the NBI/SF approach are tackled from the same starting point, a slow convergence arising from a difficult initial point persists throughout the entire solution process. On the other hand, the ROSE algorithm utilizes the user-supplied starting point only once, thereafter transitioning to initial points determined by recursively exploring the objective space. This renders its

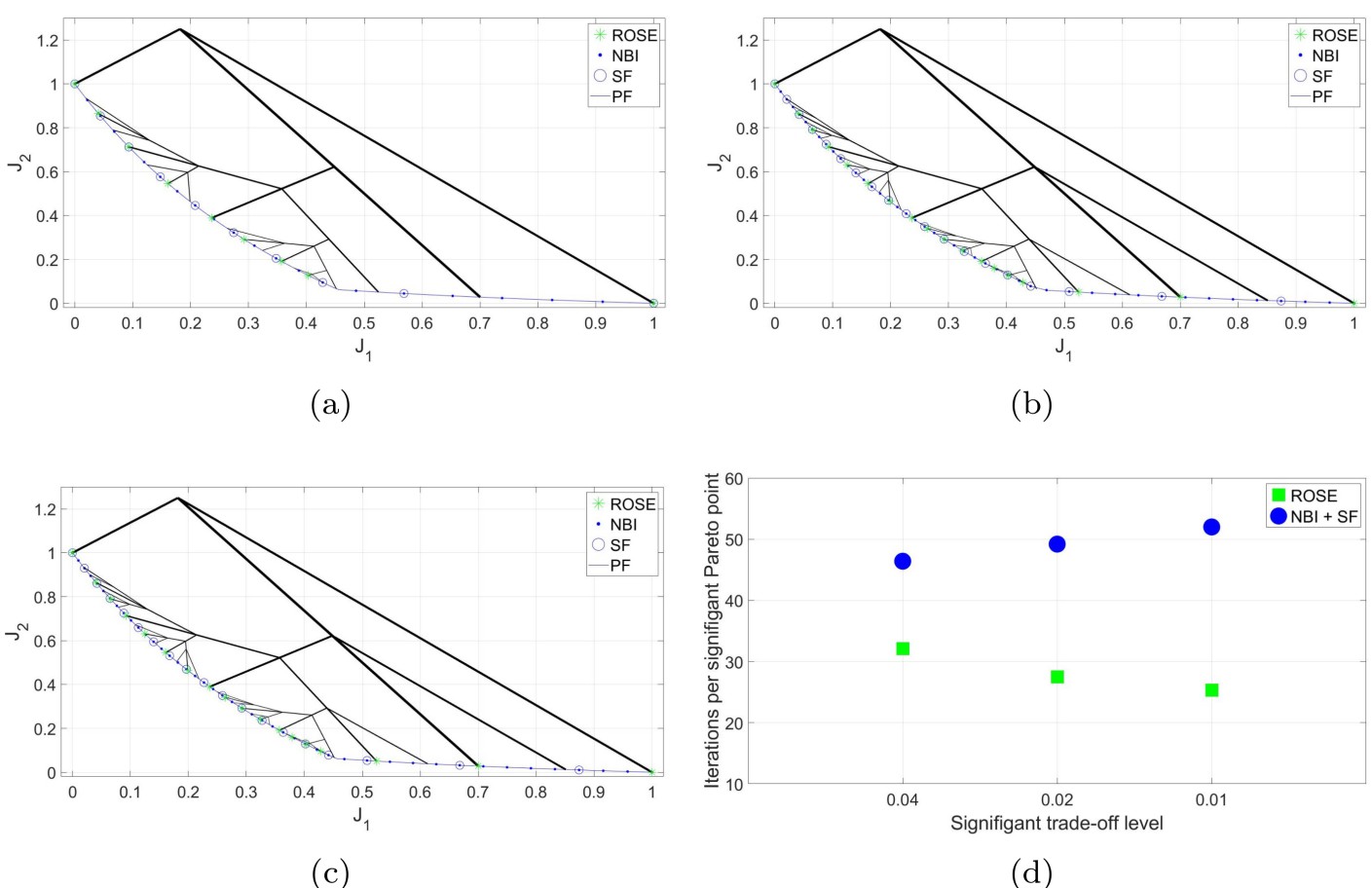

**Fig 7. Comparing ROSE and the NBI algorithm combined with a SF with varying resolutions of the ROSE stopping criterion and the SF specification when applied to the CONSTR- problem:** (a) $\Delta t = 0.04$ (b) $\Delta t = 0.02$ (c) $\Delta t = 0.01$ (d) Comparing the mean number of iterations per Pareto point for both algorithms across different ROSE stopping criterion/SF specifications.

efficiency less susceptible to adverse initial guesses. Nevertheless, it is crucial to exercise caution and not assume that the ROSE algorithm will perform equally well from every point in the objective space, despite its successful performance in the last two case studies where it started from non-central points. If the seed is located too close to the Pareto front, this can pose challenges in developing the necessary branching structure for a comprehensive exploration of the Pareto front.

**5.3.2 Extension to high-dimensional objective space.** While this paper focuses primarily on bi-objective optimization, we include here a conceptual extension to three objectives. This section is not intended as a performance benchmark, but as an exploratory demonstration of how the recursive branching scheme can be generalized to higher-dimensional objective spaces. Extending the algorithm from bi-objective optimization problems to the more general case of solving MOOP is not a straightforward task. In MOOPs, it is more common for a seed, or an intermediate node, to be in an unfavorable position within the high-dimensional Pareto surface. Hence, while the idea of extending this strategy to efficiently explore the computationally challenging high-dimensional objective space in MOOPs is appealing, innovative space-exploration strategies would need to be developed for such problems. These strategies would need to address the difficulties posed by the unfavorable starting positions and

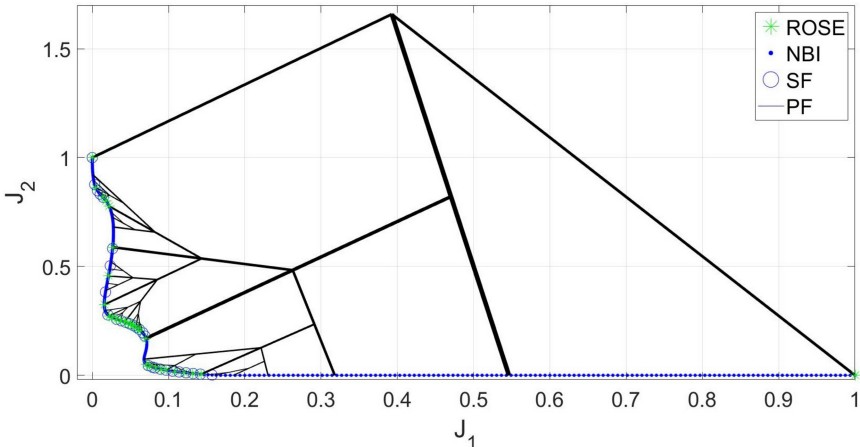

**Fig 8. Pareto front representations with adaptive density obtained by solving the DO2DK problem using the ROSE algorithm and the NBI algorithm coupled with posteriori filtering via a SF are depicted, along with a visualization of the solution process of the ROSE algorithm in the objective space. The stopping criterion of the ROSE algorithm, which aligns with the SF specification, was set to $\Delta t = 0.003$.**

the spatial complexity of high-dimensional Pareto surfaces. Nevertheless, we present a simple extension of the algorithm that has been tested on the 3 objectives problem from [36].

The algorithm begins with the initial guess $x_0$, corresponding to the seed point $(J_1(x_0), J_2(x_0), J_3(x_0))$ in the objective space. Initially, three minimization problems are solved to yield the anchor points $A_1$, $A_2$, and $A_3$. Starting from the seed point $S$ and the triangle $\triangle A_1 A_2 A_3$ in the objective space, the algorithm constructs a bisecting vector as follows:

$$\vec{B}_{A_1,A_2,A_3} = |\vec{SA_2}||\vec{SA_3}|\vec{SA_1} + |\vec{SA_1}||\vec{SA_3}|\vec{SA_2} + |\vec{SA_1}||\vec{SA_2}|\vec{SA_3} \tag{42}$$

The algorithm then solves a minimization problem along the vector $\vec{B}_{A_1,A_2,A_3}$ to find a Pareto point $P_i$, which is added to the solution set $S$. Subsequently, the closest node to the branch's center point is identified as $N_i$. Using $N_i$ and the parent triangle, the input for the next recursive call is generated through a barycentric subdivision process.

The algorithm is overviewed in Algorithm 2. The results are seen in Fig 11. It is seen that while the algorithm has successfully obtained a representation of the Pareto front, the density of the obtained points is higher at the regions closer to the seed. This reflects the geometric bias introduced by triangle subdivision near the seed. Nevertheless, the recursive branching structure generalizes naturally to higher dimensions. Regarding stopping criteria in multi-objective optimization, possible strategies include direct trade-off comparisons between generated points, centroid-based relevance tests, and information-theoretic measures based on local objective variance [20]. Two key challenges for future extensions are ensuring uniform exploration of high-dimensional trade-off surfaces and mitigating sensitivity to the initial seed placement, which becomes increasingly critical as the number of objectives increases. The formulation of the general N-dimensional algorithm is provided in S1 Algorithm.

## 6 Conclusion

In this paper, inspired by branching phenomena in nature as an efficient means to explore physical space, we introduce an algorithm that utilizes a branching strategy to explore the

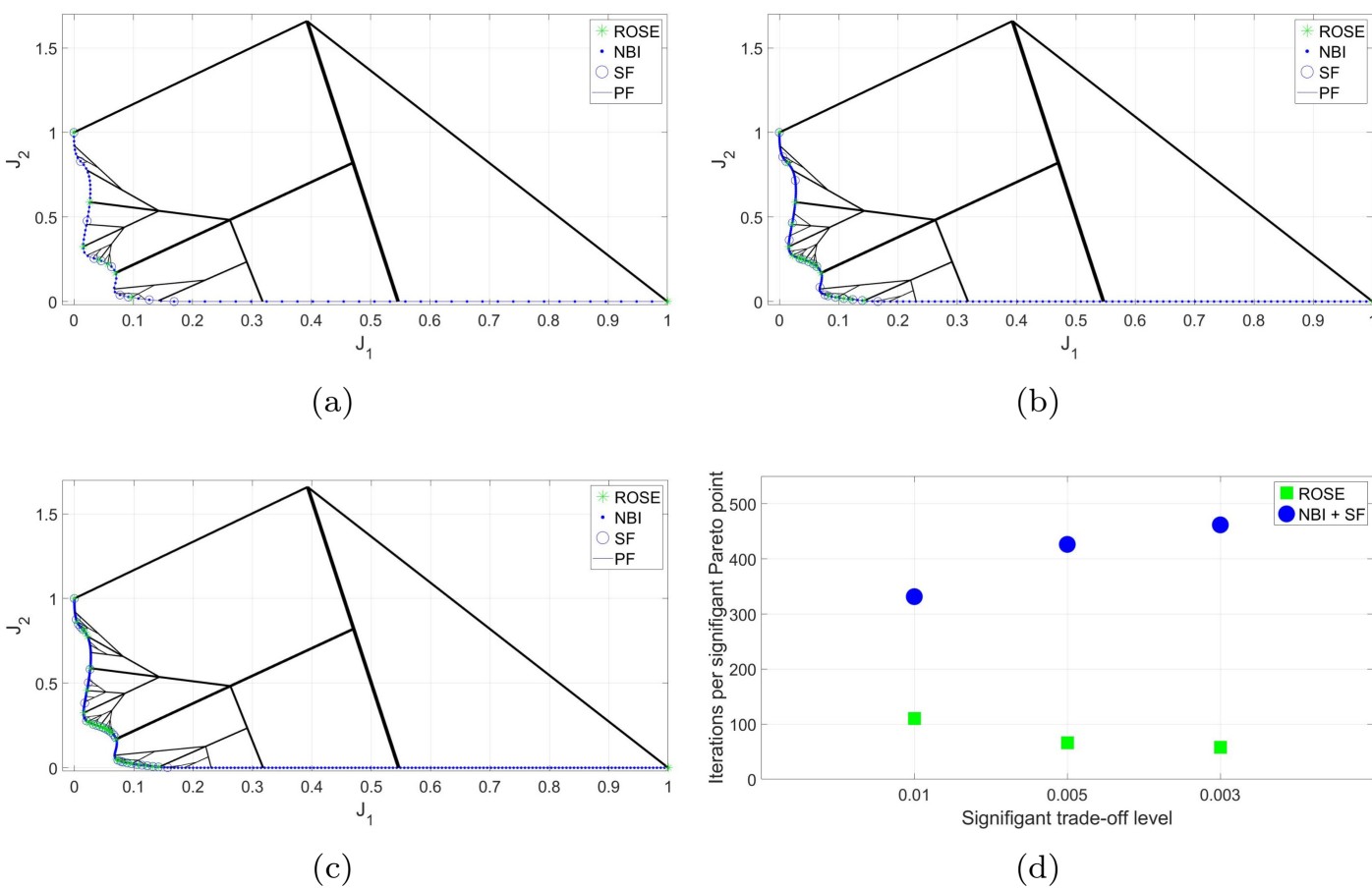

**Fig 9. Comparing ROSE and the NBI algorithm combined with a SF with varying resolutions of the ROSE stopping criterion and the SF specification when applied to the DOD2K problem:** (a) $\Delta t = 0.01$ (b) $\Delta t = 0.005$ (c) $\Delta t = 0.003$ (d) Comparing the mean number of iterations per Pareto point for both algorithms across different ROSE stopping criterion/SF specifications.

objective space in bi-objective optimization problems. This occurs via solving a series of intermediate problems that structure the solution process in the objective space in a recursive manner, allowing efficient exploration where no region of the objective space gets explored twice. The behavior of the ROSE algorithm is primarily governed by two parameters: the initial guess $x_0$, which determines the seed point in the objective space, and the trade-off threshold $\Delta t$, which sets the resolution of the Pareto set. While $x_0$ initiates the exploration, the recursive structure shifts focus toward regions of higher trade-offs, making the algorithm relatively robust to the initial seed, as seen across varied runs (see S1 Text). In contrast, $\Delta t$ directly controls the density of the resulting Pareto front. Smaller values of $\Delta t$ yield finer approximations in information-rich regions and enable early termination of growth in flat areas, thereby improving efficiency.

The ROSE algorithm's performance has been illustrated via a number of case studies and benchmarked against a standard strategy from literature to obtain a Pareto front with adaptive resolution, namely obtaining a dense Pareto front first with NBI then subjecting it to posteriori filtering via a SF. The presented approach has two major advantages over current scalarization strategies. First, the ROSE algorithm showed higher computational efficiency both for generating a Pareto front with the same number of points as NBI, where the gain in speed comes from the more efficient exploration of the objective space, and also when producing a

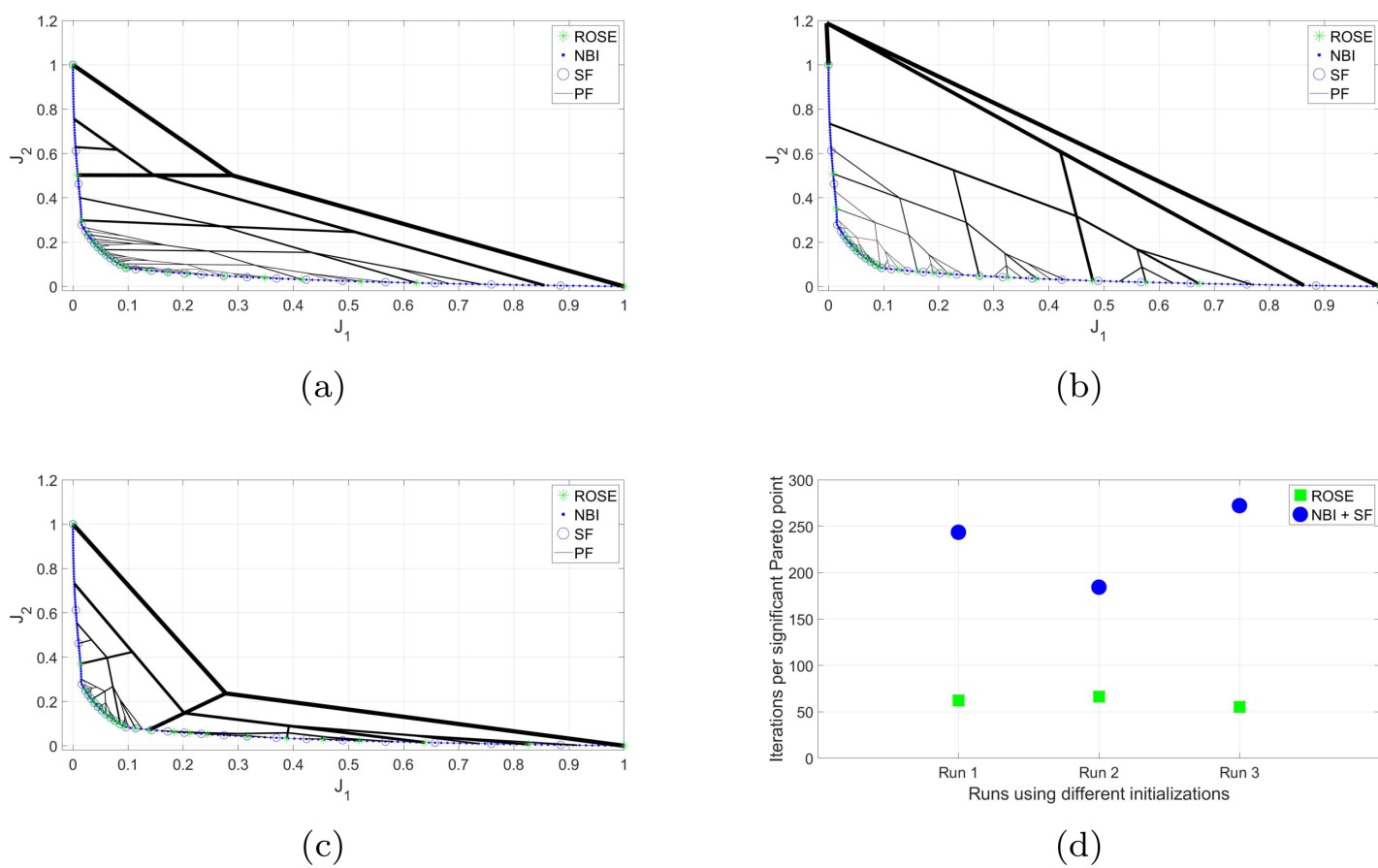

**Fig 10. Comparing ROSE and the NBI algorithm combined with a SF with same** $\Delta t = 0.005$**, applied to the problem of the bi-objective optimization of an I-Beam using different initial values of** $x_o$ **that correspond to starting at different points in the objective space: (a) Run 1,** $x_o = [45, 35, 3.8, 1.2]$ **(b)** $x_o = [30, 20, 1.5, 4]$ **(c)** $x_o = [45, 35, 2, 4]$ **(d) Comparing the mean number of iterations per significant Pareto point for both algorithms across runs from different initial guesses.**

Pareto front with adaptive resolution where a further gain of speed comes from the ability of terminating the solution process early at areas of the Pareto front that are not interesting to the DM. The second advantage is having a clear, intuitive and DM oriented stopping criterion when compared to traditional approaches. That being said, while the algorithm has shown some flexibility with respect to the seed position in the objective space, there are challenges to generalize the algorithm to the MOOPs, since it still requires the seed of the solution process to be in a favorable location, relatively far from the Pareto front, to allow a comprehensive exploration of the objective space.

Our choice of benchmarking the ROSE algorithm against NBI is grounded in two key reasons. Firstly, NBI serves as a typical representative of the scalarization class of algorithms, sharing similarities with methods like the epsilon-constraint approach, which also discretize a MOOP into a set of SOOPs, leading to redundant exploration of the objective space. This similarity extends to the weighted sum method, which not only suffers from repetitive exploration but also struggles to uniformly represent the Pareto front. Second, the simple parameterization of the NBI/SF strategy, which is one of its notable strengths, matches the simplicity of our algorithm's parameter setup. This contrasts with other algorithms, which, although

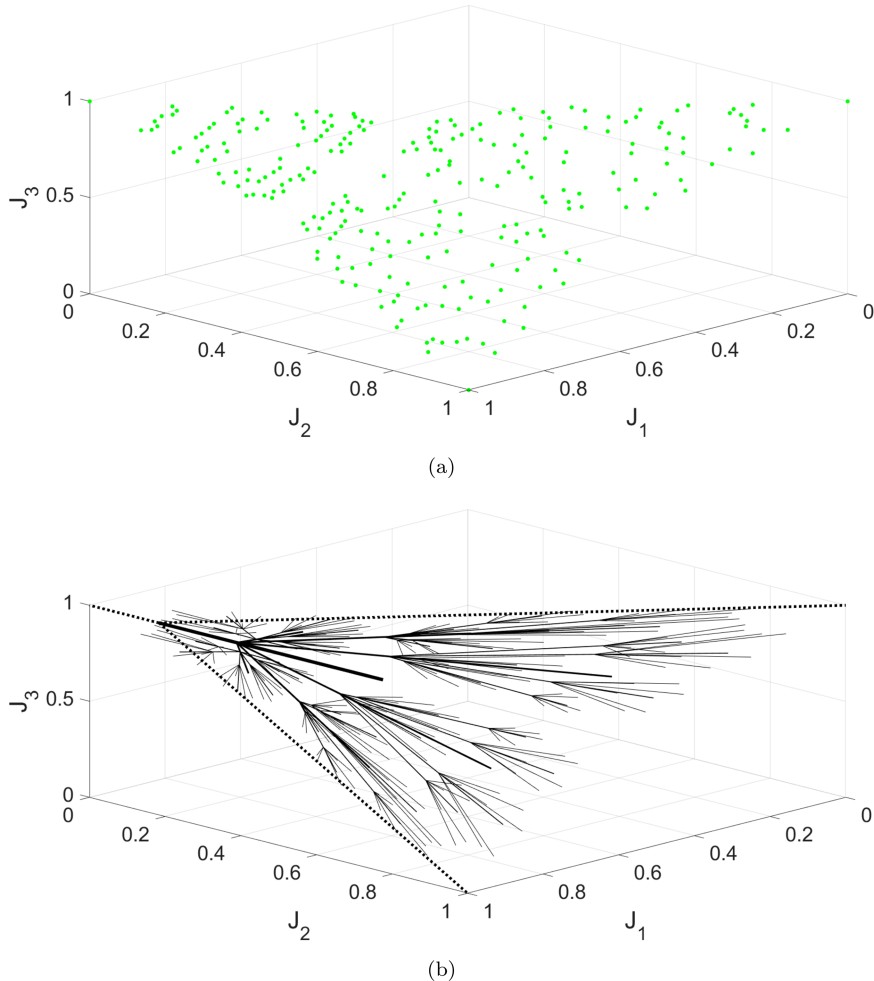

**Fig 11. Solving the three-objectives numerical problem from [36] using a generalized version of the ROSE algorithm for three-objective problems: (a) The Pareto front representation corresponding to three recursive levels of the algorithm. (b) Visualization of the solution process in the three-dimensional objective space.**

potentially more efficient with optimal parameterization, require a more extensive set of parameters [6,41].

It is also noteworthy that the ROSE algorithm can be viewed as a structured approach for warm starting the solution process. In comparison to other warm start techniques like Bayesian optimization or quasi-Monte Carlo, which are efficient when optimally parametrized but necessitate parameter tuning, familiarity with statistical methods, and problem-specific knowledge, ROSE requires simple parameterization. It only necessitates an initial guess and a decision-maker oriented stopping criterion, thus widening its applicability. Furthermore, while this work focuses on deterministic scalarization, it is worth briefly contrasting ROSE with population-based vectorization methods such as MOEA/D [46] and NSGA-II [13]. These approaches evolve a population of solutions using genetic operators or decomposition strategies. In contrast, ROSE constructs each Pareto point via a directed, recursive search in objective space, offering greater transparency, warm-start potential, and DM-oriented resolution control via $\Delta t$. Although population-based methods remain powerful for many-objective

**Algorithm 2 Recursive objective space exploration in 3D via barycentric subdivision.**

```
Input: Initial guess x_o, and a predefined stopping criterion.
```

```
Output: Pareto set S.
```

***Step 1:*** `Initialize the solution set` $S = \{\}$`.`

***Step 2:*** `Generate the seed point based on` $x_o$`:` $(J_1(x_o), J_2(x_o), J_3(x_o))$

***Step 3:*** `Solve` $\min_x\{J_1\}$`,` $\min_x\{J_2\}$`, and` $\min_x\{J_3\}$ `to get anchor points` $A_1$`,` $A_2$`, and` $A_3$`.`

***Step 4:*** `Start the recursive process with the branching node` $S$ `and triangle` $\triangle A_1 A_2 A_3$`.`

`Construct the bisecting vector` $\vec{B}_{A_1,A_2,A_3}$ `as:`

$$\vec{B}_{A_1,A_2,A_3} = |\vec{SA_2}||\vec{SA_3}|\vec{SA_1} + |\vec{SA_1}||\vec{SA_3}|\vec{SA_2} + |\vec{SA_1}||\vec{SA_2}|\vec{SA_3}$$

`Solve sub-problem` $\min_x -|\vec{B}_{A_1,A_2,A_3}|$ `to find Pareto point` $P_i$`.`

`if the stopping criterion is not activated` **then**

`Add` $P_i$ `to` $S$

`Find the node` $N_i$ `closest to the branch's center point.`

`Divide the parent triangle into 6 smaller triangles using barycentric subdivision, feed each triangle back to the recursive function along with node` $N_i$`.`

***Step 5:*** `When all recursive calls are exited, produce the solution set` $S$`.`

and multimodal problems, benchmarking ROSE against them is an important avenue for future work.

To conclude, this algorithm demonstrates how exploring the objective space more efficiently can enhance the solving process of MOOPs. Most existing methods for MOOPs rely on scalarization techniques, which convert the problem into a set of SOOPs that are solved independently. Our approach shows the potential benefits, both computational and conceptual, of addressing the MOOP as a whole, by organizing the solution process within the objective space. While we have focused on bi-objective optimization, our approach can be extended to more general MOOPs, which are notoriously computationally demanding. However, technical challenges have to be overcome first by devising novel, more robust, high dimensional space-exploration strategies. Furthermore, another research direction that is urgently needed is to design optimization solvers that are specifically tailored for finding the Pareto front instead of using solvers that are optimized for SOOPs. Such ideas would open up future avenues of work that, in addition to being more computationally efficient, would further reveal the mathematical beauty of solving MOOPs.

## Supporting information

**S1 File. ROSE Supplementary Material. This file contains supplementary analysis and figures demonstrating the robustness of the ROSE algorithm to different initial seed points in the bi-objective benchmark problem, as well as pseudo-code for the N-dimensional generalization of the ROSE algorithm.**
(PDF)

## Acknowledgments

We thank Lydia Katsini for assistance with editing the manuscript.

## Author contributions

**Conceptualization:** Ihab Hashem, Viviane De Buck, Seppe Seghers, Jan Van Impe.

**Formal analysis:** Ihab Hashem.

**Funding acquisition:** Jan Van Impe.

**Methodology:** Ihab Hashem, Viviane De Buck.

**Software:** Ihab Hashem, Seppe Seghers.

**Writing – original draft:** Ihab Hashem.

**Writing – review & editing:** Ihab Hashem, Viviane De Buck, Jan Van Impe.

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
