## [Decision Letter · Decision Letter 0]

15 Apr 2025

PONE-D-25-13070

Recursive Objective Space Exploration (ROSE): A Computationally Efficient Deterministic Approach for Bi-Objective Optimization.

PLOS ONE

Dear Dr. Van Impe,

Thank you for submitting your manuscript to PLOS ONE. After careful consideration, we feel that it has merit but does not fully meet PLOS ONE’s publication criteria as it currently stands. Therefore, we invite you to submit a revised version of the manuscript that addresses the points raised during the review process.

We look forward to receiving your revised manuscript.

Kind regards,

Yuanchao Liu

Academic Editor

PLOS ONE

**Comments from PLOS Editorial Office**: We note that one or more reviewers has recommended that you cite specific previously published works in the current and previous rounds of revision. As always, we recommend that you please review and evaluate the requested works to determine whether they are relevant and should be cited. It is not a requirement to cite these works and you may remove any added citations before the manuscript proceeds to publication. We appreciate your attention to this request.

**Journal Requirements**:

4. Please note that funding information should not appear in any section or other areas of your manuscript. We will only publish funding information present in the Funding Statement section of the online submission form. Please remove any funding-related text from the manuscript.

5. Please note that your Data Availability Statement is currently missing the repository name and/or the DOI/accession number of each dataset OR a direct link to access each database. If your manuscript is accepted for publication, you will be asked to provide these details on a very short timeline. We therefore suggest that you provide this information now, though we will not hold up the peer review process if you are unable.

Reviewers' comments:

Reviewer's Responses to Questions

**Comments to the Author**

1. Is the manuscript technically sound, and do the data support the conclusions?

Reviewer #1: Yes

Reviewer #2: Yes

2. Has the statistical analysis been performed appropriately and rigorously? 

Reviewer #1: No

Reviewer #2: Yes

3. Have the authors made all data underlying the findings in their manuscript fully available?

Reviewer #1: No

Reviewer #2: No

4. Is the manuscript presented in an intelligible fashion and written in standard English?

Reviewer #1: Yes

Reviewer #2: Yes

5. Review Comments to the Author

Reviewer #1: A/ Although an extension to tri-objective problems is proposed, it’s not comprehensively benchmarked or validated. Only a small, simple 3-objective problem is tested.

B/ There is no formal computational complexity analysis of ROSE (e.g., O(n) or O(log n) behavior), which would help quantify its advantage over NBI/SF more rigorously.

C/ The sensitivity of performance to the initial guess (seed) is discussed but not thoroughly investigated with varying objective landscapes or seed placements.

D/ A few typos and grammar inconsistencies exist (e.g., “lead” vs. “led”, “thesef weights”), but they don’t significantly impact comprehension.

E/ Briefly compare ROSE with vectorization approaches (e.g., MOEA/D, NSGA-II) in discussion, even if not tested. Include pseudo-code in the appendix for the high-dimensional version.

F/ Add some recent works that enhance various objectives and applications related to optimizations in your work as below:

- https://doi.org/10.1016/j.engappai.2024.108980

- https://doi.org/10.1016/j.engappai.2024.108053

- https://doi.org/10.1109/JSEN.2024.3477960

Reviewer #2: This work proposes an alternative strategy to tackle bi-objective optimization problems by exploring the objective space recursively at a reduced computational cost. Although this work has certain innovations, there are some problems that need to address:

1) All the figures cannot find in the manuscript.

2) The motivations of developing this algorithm should be clearly given.

3) Authors should give an intuitive illustration of recursive objective space exploration by a figure.

4) Authors should conduct the ablation experiments to validate the importance of each main components.

5) There are some expensive multi-objective optimization algorithms that have been developed to reduce the computational cost[1-3]. Authors can authors can read these references to increase the depth of this work.

[1] Liu Y, Liu J, Jin Y. Surrogate-assisted multipopulation particle swarm optimizer for high-dimensional expensive optimization[J]. IEEE Transactions on Systems, Man, and Cybernetics: Systems, 2021, 52(7): 4671-4684.

[2] Lv Z, Wang L, Han Z, et al. Surrogate-assisted particle swarm optimization algorithm with Pareto active learning for expensive multi-objective optimization[J]. IEEE/CAA Journal of Automatica Sinica, 2019, 6(3): 838-849.

[3] Song Z, Wang H, Xue B, et al. Balancing objective optimization and constraint satisfaction in expensive constrained evolutionary multi-objective optimization[J]. IEEE Transactions on Evolutionary Computation, 2023.

6) Please check the paper carefully, there are many typos in the manuscript.

6. PLOS authors have the option to publish the peer review history of their article (what does this mean?). If published, this will include your full peer review and any attached files.

Reviewer #1: No

Reviewer #2: No

---

## [Author Response · Author response to Decision Letter 1]

29 May 2025

See the Response to Reviews file.

---

## [Decision Letter · Decision Letter 1]

25 Jun 2025

Recursive Objective Space Exploration (ROSE): A Computationally Efficient Deterministic Approach for Bi-Objective Optimization.

PONE-D-25-13070R1

Dear Dr. Jan F M Van Impe,

We’re pleased to inform you that your manuscript has been judged scientifically suitable for publication and will be formally accepted for publication once it meets all outstanding technical requirements.

Kind regards,

Yuanchao Liu

Academic Editor

PLOS ONE

Additional Editor Comments (optional):

Reviewers' comments:

Reviewer's Responses to Questions

**Comments to the Author**

1. If the authors have adequately addressed your comments raised in a previous round of review and you feel that this manuscript is now acceptable for publication, you may indicate that here to bypass the “Comments to the Author” section, enter your conflict of interest statement in the “Confidential to Editor” section, and submit your "Accept" recommendation.

Reviewer #2: All comments have been addressed

Reviewer #3: All comments have been addressed

2. Is the manuscript technically sound, and do the data support the conclusions?

Reviewer #2: Yes

Reviewer #3: Yes

3. Has the statistical analysis been performed appropriately and rigorously? 

Reviewer #2: Yes

Reviewer #3: Yes

4. Have the authors made all data underlying the findings in their manuscript fully available?

Reviewer #2: Yes

Reviewer #3: Yes

5. Is the manuscript presented in an intelligible fashion and written in standard English?

Reviewer #2: Yes

Reviewer #3: Yes

6. Review Comments to the Author

Reviewer #2: Authors have carefully addressed my comments. Therefore, this manuscript can be accepted to publish.

Reviewer #3: I have carefully reviewed the revised version of the manuscript. The authors have addressed all the concerns raised in the previous round of review, and the modifications made are appropriate and well executed. The revised manuscript demonstrates improved clarity, technical rigor, and structural coherence. I find the current version acceptable and recommend it for publication in its present form.

7. PLOS authors have the option to publish the peer review history of their article (what does this mean?). If published, this will include your full peer review and any attached files.

Reviewer #2: No

Reviewer #3: No

---

## [Editor Report · Acceptance letter]

PONE-D-25-13070R1

PLOS ONE

Dear Dr. Impe,

I'm pleased to inform you that your manuscript has been deemed suitable for publication in PLOS ONE. Congratulations! Your manuscript is now being handed over to our production team.

Kind regards,

on behalf of

Dr. Yuanchao Liu

Academic Editor

PLOS ONE